# Cathepsin B Protease Facilitates Chikungunya Virus Envelope Protein-Mediated Infection Via Endocytosis or Macropinocytosis

**DOI:** 10.3390/v12070722

**Published:** 2020-07-03

**Authors:** Mai Izumida, Hideki Hayashi, Atsushi Tanaka, Yoshinao Kubo

**Affiliations:** 1Department of Clinical Medicine, Institute of Tropical Medicine, Nagasaki University, Nagasaki 852-8523, Japan; 2Medical University Research Administrator, Nagasaki University School of Medicine, Nagasaki 852-8523, Japan; hhayashi@nagasaki-u.ac.jp; 3Research Institute for Microbial Diseases, Osaka University, Osaka 565-0871, Japan; washi69@biken.osaka-u.ac.jp; 4Program for Nurturing Global Leaders in Tropical and Emerging Communicable Diseases, Graduate School of Biomedical Sciences, Nagasaki University, Nagasaki 852-8523, Japan

**Keywords:** chikungunya virus, murine leukemia virus vector, endocytosis, macropinocytosis, cathepsin B

## Abstract

Chikungunya virus (CHIKV) is an enveloped virus that enters host cells and transits within the endosomes before starting its replication cycle, the precise mechanism of which is yet to be elucidated. Endocytosis and endosome acidification inhibitors inhibit infection by CHIKV, murine leukemia virus (MLV), or SARS-coronavirus, indicating that these viral entries into host cells occur through endosomes and require endosome acidification. Although endosomal cathepsin B protease is necessary for MLV, Ebola virus, and SARS-CoV infections, its role in CHIKV infection is unknown. Our results revealed that endocytosis inhibitors attenuated CHIKV-pseudotyped MLV vector infection in 293T cells but not in TE671 cells. In contrast, macropinocytosis inhibitors attenuated CHIKV-pseudotyped MLV vector infection in TE671 cells but not in 293T cells, suggesting that CHIKV host cell entry occurs via endocytosis or macropinocytosis, depending on the cell lines used. Cathepsin B inhibitor and knockdown by an shRNA suppressed CHIKV-pseudotyped MLV vector infection both in 293T and TE671 cells. These results show that cathepsin B facilitates CHIKV infection regardless of the entry pathway.

## 1. Introduction

Human infection by the mosquito-borne chikungunya virus (CHIKV) induces fever frequently accompanied by severe joint pain, muscle pain, headache, nausea, fatigue, and rash [1]. Chikungunya fever mainly occurs in Africa, and South and Southeast Asia [1]. However, effective therapeutic drugs are not available at present. Chikungunya virus is a member of the genus *Alphavirus* in the family *Togaviridae* and has an envelope membrane, as do HIV, Ebola virus, and severe acute respiratory syndrome coronavirus (SARS-CoV). The molecular mechanism of CHIKV entry into human target cells has not been studied to the same extent as HIV, Ebola virus, and SARS-CoV.

The binding of CHIKV to a cognate cell surface receptor initiates CHIKV infection. Prohibition [2], TIM-1 [3], glycosaminoglycans [4], heparan sulfate [5], and ATP synthase β subunit [6] have been reported to participate in the CHIKV infection process. However, CHIKV infection can occur in the absence of these proteins, indicating that they facilitate the initial binding of CHIKV to host cells. Mxra8 was recently reported as the cell surface receptor for CHIKV [7].

There are many lines of evidence showing that CHIKV enters into host cells via clathrin-dependent endocytosis pathway [8,9,10,11]. The inhibitors of endosome acidification attenuate CHIKV infection [12,13,14]. Therefore, it is thought that CHIKV particles are internalized into host cell endosomes after binding to the cell surface receptor, and then endosome acidification facilitates fusion between viral and endosome membrane by the CHIKV envelope (E) protein. In addition, it has been reported that phogocytosis inhibitors are able to reduce CHIKV infection [15] and antibody-bound viral particles can follow an Fcγ-dependent phagocytosis pathway leading to antibody-dependent enhancement of viral infection [16], suggesting that CHIKV also enters to target cells via phagosomes.

CHIKV entry into target cells is mediated by the viral E protein. The viral protein is synthesized as a precursor polyprotein from a subgenomic 26s RNA and processed into capsid, E3, E2, 6K, and E1 mature proteins. Spikes on the CHIKV particles consist of trimers of heterodimers containing E1 and E2 proteins. The E proteins that emerge from the viral envelope play an important role for attachment to the cell receptor and entry step for many cell types, but not for all. The E proteins may also be involved in the post-entry step.

Infections by Ebola virus [17,18], SARS-CoV [19], SARS-CoV-2 [20], and murine leukemia virus (MLV) [18] also require a transit via the endosome compartment, and are endosome acidification-dependent. Furthermore, these viral infections require endosomal cathepsin B protease [21,22,23,24], suggesting that the cleavage of their envelope glycoproteins by cathepsin B protease activates their membrane fusion capability. However, the involvement of cathepsin B protease in CHIKV infection remains to be elucidated.

In this study, we examined whether cathepsin B is important for efficient CHIKV infection in 293T, HeLa, and TE671 cells. 293T and HeLa cells are widely used as target cells for in vitro CHIKV infection [25,26,27,28,29,30,31]. TE671 rhabdomyosarcoma cell line was also used, as skeletal muscle cells have found to contain CHIKV antigens [32]. Cathepsin B protease is required for uncoating of adeno-associated virus core [33]. We used CHIKV E protein-containing MLV vector to examine only the interaction between cathepsin B and CHIKV E proteins, and to exclude the involvement of cathepsin B in viral uncoating. Such pseudotyped MLV vector is widely used to understand the mechanism of cell entry mediated by various viral glycoproteins [34].

## 2. Materials and Methods

### 2.1. Cells

Human 293T, TE671, and HeLa cell lines have been maintained in our laboratory for a long period. They were cultured in Dulbecco’s modified Eagle’s medium with 8% fetal bovine serum and 1% penicillin-streptomycin.

### 2.2. Plasmids

The MLV Gag-Pol expression plasmid was purchased from TaKaRa. The construction of a LacZ-encoding MLV vector genome expression plasmid has been previously reported [35]. The VSV-G expression plasmid was kindly provided by Dr. L. Chang [36]. The CHIKV E protein expression plasmid was constructed from an artificially synthesized human codon-optimized full-length DNA fragment encoding CHIKV E protein [5] (Eurofins Genomics) that was cloned into a pT_ARGE_T™ plasmid (Promega, Madison, WI, USA).

### 2.3. Pseudotyped MLV Vector

293T cells were transfected by the MLV Gag-Pol (TaKaRa, Shiga, Japan), LacZ-encoding MLV vector genome, and appropriate viral envelope protein expression plasmids using Fugene^®^ HD Transfection Reagent (Promega) (Appendix A). The culture medium was refreshed 24 h following transfection, and the cells were cultured for an additional 24 h. Then, the culture medium was centrifuged to remove cells and cell debris. For the CHIKV-pseudotyped MLV vector, the target cells were incubated with the undiluted supernatant. For the VSV-pseudotyped MLV vector, the supernatant was diluted 1/10 with fresh medium, and target cells were inoculated with the diluted supernatant. The inoculated cells were cultured for 2 days and then stained with X-Gal (Wako Pure Chemical Industries, Osaka, Japan) for 5 h. The stained blue cells were counted in eight randomly selected microscope fields using a 10 × objective glass in an experiment. Usually, 10–50 blue cells were detected per microscope field. The total numbers of blue cells in the eight microscope fields were compared. This experiment was independently repeated three times. Numbers of infected blue cells were much lower than those of total target cells (MOI < 1%). As transduction efficiency was changed day-by-day, the control and treated cells were incubated with the same culture supernatant from the transfected cells.

### 2.4. Cell Viability

The target cells were treated with the indicated inhibitors for 5 h, and were subsequently cultured for an additional 24 h in fresh medium. The control cells were treated with the same volume of dimethyl sulfoxide (DMSO). The treated cells were stained with trypan blue, and the numbers of unstained (viable) cells were counted using a counting chamber.

### 2.5. Macropinocytosis Assay

The target cells were treated with the indicated inhibitors, and the control cells were treated with the same volume of DMSO for 5 h. The treated cells were then cultured for an additional 5 h in the presence of FITC-conjugated dextran (molecular weight >70,000) (Life Technologies Corporation, Eugene, OR, USA). The cells were washed with phosphate-buffered saline and treated with 1% acetic acid in phosphate-buffered saline to remove FITC-conjugated dextran bound to the cell surfaces. Means of fluorescence intensities (MFIs) of the cells were measured using a flow cytometer (BD Biosciences, Franklin Lakes, NJ, USA).

### 2.6. Mouse Antiserum against CHIKV E2 Protein

Antiserum against CHIKV E2 protein was produced by Sigma-Aldrich. Briefly, a peptide derived from CHIKV E2 protein (Appendix A), was inoculated into a rabbit several times, and the serum was isolated from the rabbit. The amino acid sequence of the antigen peptide was PVIGRERFHSRPOHGKELPC.

### 2.7. Western Immunoblotting

Cell lysates were prepared from 293T cells transfected by the CHIKV-pseudotyped MLV vector construction plasmids. Culture supernatants were centrifuged at 1000 rpm for 10 min to remove cells and cell debris. The MLV vector particles were collected by centrifugation of the supernatants at 15,000 rpm through 20% sucrose. The cell lysates and virion pellets were subjected to SDS-PAGE. The proteins were transferred onto polyvinylidene difluoride membranes. The membranes were treated with the rabbit anti-CHIKV E2 antiserum and goat anti-MLV p30 antibody (ViroMed Laboratories, Burlington, North Carolina, USA) and then with HRP-conjugated antirabbit and antigoat IgG antibodies (Bio-Rad, Herules, California, USA), respectively. The antibody-bound proteins were visualized using Clarity Western ECL Substrate (Bio-Rad).

### 2.8. RT-PCR

Total RNA was isolated using TRIzol™ reagent (Thermo Fisher Scientific, Carlsbad, California, USA). First strand cDNA was synthesized with ProtoScript^®^ II Reverse Transcriptase (New England BioLabs, Ipswich, Massachusetts, USA) using random hexamers (TaKaRa). Glyceraldehyde-3-phosphate dehydrogenase (control) or cathepsin B sequences were amplified by PCR.

### 2.9. Statistical Analysis

Differences between two groups were determined by the Student’s *t*-test. The difference was considered statistically significant if the *p*-value was <0.05 for all tests.

## 3. Results

### 3.1. Characterization of CHIKV-Pseudotyped Retrovirus Vector

We used a CHIKV-pseudotyped MLV vector to measure the CHIKV E protein-mediated entry into target cells. To construct the CHIKV-pseudotyped MLV vector, a full-length DNA fragment containing the CHIKV E3-E2-6K-E1 sequence optimized to human codon usage was synthesized and inserted into a pT_ARGE_T™ expression plasmid. 293T cells were transfected with the CHIKV E protein expression plasmid together with MLV Gag-Pol and LacZ-encoding MLV vector genome expression plasmids (Appendix A). Human 293T, HeLa, or TE671 cells were incubated with the culture supernatants of the transfected cells. The inoculated cells were stained with X-Gal, and the resultant blue cells were counted to measure the CHIKV-pseudotyped MLV vector infection. 293T and TE671 cells were used in the subsequent experiments, because the numbers of blue 293T and TE671 cells were much higher than those of the blue HeLa cells (Figure 1A). Only a few infected cells were detected in HeLa cells.

We wanted to verify that the infection by the pseudotyped MLV vector is achieved by a recognition step dependent on CHIKV E protein and is neutralized by an antiserum, as replication-competent CHIKV. The effect of a serum that was isolated from CHIKV-infected patients and has neutralization activity against CHIKV infection [37,38,39] was analyzed. Human embryonic kidney 293T or rhabdomyosarcoma TE671 cells were incubated with the CHIKV E protein-containing MLV vector in the presence of the serum. Fifty percent focus reduction neutralization value of the serum was 1:2560 (personal communication with Dr. Inoue). Similarly, when the serum was diluted with medium by 1/2000, infected cell numbers were decreased to about 50% (Figure 1B). However, the serum less efficiently inhibited vesicular stomatitis virus (VSV)-pseudotyped MLV vector infection than CHIKV-pseudotyped vector. This result confirmed that the infection is induced by the CHIKV E protein.

To examine whether CHIKV-pseudotyped MLV vector infection required endosome acidification, 293T or TE671 cells were pretreated with the endosome acidification inhibitor concanamycin A (CMA) [40] for 5 h. This CMA treatment did not significantly suppress the cell viability [18]. The treated cells were incubated with the CHIKV-pseudotyped MLV vector. The CMA treatment inhibited the CHIKV-pseudotyped MLV vector infection in a dose-dependent manner (Figure 1C), which confirmed that endosome acidification was required for CHIKV-pseudotyped vector infection, as previously reported [12,13,14].

### 3.2. Endocytosis Is Required for CHIKV-Pseudotyped MLV Vector Infection in 293T but Not in TE671 Cells

The above result prompted us to speculate that CHIKV-pseudotyped MLV vector infection occurs via endocytosis in 293T and TE671 cells. To this end, 293T and TE671 cells were pretreated with an endocytosis inhibitor, dynasore (20 or 40 μM) or Pitstop2 (50 or 100 μM) for 5 h. Dynasore inhibits vesicle formation [41]. Pitstop2 is a specific inhibitor of clathrin that suppresses clathrin-mediated endocytosis [42]. We have already reported that the dynasore treatment inhibits ecotropic MLV infection without affecting the cell viability [18]. The Pitstop2 treatment at 50 μM did not affect cell viability, and that at 100 μM decreased live cell numbers to about 60% (Figure 2A). The treated cells were incubated with a CHIKV- or VSV-pseudotyped MLV vector. Dynasore (Figure 2B) and Pitstop2 (Figure 2C) both inhibited the CHIKV-pseudotyped MLV vector infection in 293T cells but not in TE671 cells. These inhibitors more efficiently decreased the infected cell numbers than the live cell numbers in 293T cells. However, the VSV-pseudotyped MLV vector infection was attenuated by the inhibitors in both 293T and TE671 cells (Figure 2D,E). The VSV-pseudotyped retroviral vectors are widely utilized as an endocytic model of virus infection. Endocytosis would have been suppressed in the treated cells, because the VSV-pseudotyped MLV vector infection was inhibited by dynasore and Pitstop2. These results suggested that the CHIKV-pseudotyped vector infection required endocytosis in 293T cells but not in TE671 cells.

### 3.3. Macropinocytosis Is Required for CHIKV-Pseudotyped MLV Vector Infection in TE671 Cells but Not in 293T Cells

The endocytosis inhibitors did not suppress the CHIKV-pseudotyped MLV vector infection in TE671 cells. To assess whether macropinocytosis was the mechanism of infection in TE671 cells, 293T and TE671 cells were pretreated with the macropinocytosis inhibitors 5-(N-ethyl-N-isopropyl)-amiloride (EIPA) (10 or 20 μM) for 5 h. TE671 cells were also treated with wortmannin (WOR) at 50 or 100 μM, and 293T cells were at 10 or 20 μM, because the WOR treatment at 50 and 100 μM had severe cytotoxity in 293T cells. EIPA inhibits the Na^+^/H^+^ ion exchange pump in the plasma membrane, thereby affecting the intracellular pH, resulting in the cessation of macropinocytosis [43]. WOR specifically inhibits PI3-kinase, which is essential for macropinocytosis [44]. EIAP treatment only slightly suppressed the cell viability, but not WOR treatment (Figure 3A,B).

To examine whether the EIPA or WOR treatment inhibits macropinocytosis, fluorescein isothiocyanate (FITC)-labeled dextran (molecular weight >70,000) was added to the treated cells. The dextran could be internalized by macropinocytosis, but not by endocytosis, because of its large molecular size. 293T and TE671 cells were treated with EIPA at 20 μM. 293T and TE671 cells were treated with WOR at 20 and 100 μM, respectively. The fluorescence intensities of the 293T and TE671 cells were decreased by the EIPA or WOR treatment (Figure 3C). These results demonstrated that macropinocytosis was indeed suppressed by these inhibitors.

Then, the effect of the macripinocytosis inhibitors on CHIKV-pseudotyped MLV vector infection was analyzed. Both macropinocytosis inhibitors attenuated the CHIKV-pseudotyped MLV vector infection in TE671 cells in a dose-dependent manner but not in 293T cells (Figure 4A,B). When TE671 cells were treated with 10 μM EIPA, the CHIKV-pseudotyped MLV infection was inhibited (Figure 4A) but the VSV-pseudotyped infection was not (Figure 4C). The EIPA treatment at 20 μM more significantly inhibited the CHIKV-pseudotyped MLV infection than VSV-pseudotyped infection. The EIPA treatment inhibited VSV-pseudotyped MLV vector infection in TE671 cells, but not in 293T cells (Figure 4C). The WOR treatment did not affect the VSV-pseudotyped vector infection in 293T cells, and rather enhanced the infection in TE671 cells (Figure 4D). These results show that CHIKV-pseudotyped vector infection occurs through macropinocytosis in rhabdomyosarcoma TE671 cells, as previously reported in another rhabdomyosarcoma cell line, SJCRH30 [45].

To confirm whether the macropinocytosis inhibitors suppress virus entry, target 293T and TE671 cells were first inoculated with CHIKV-pseudotyped MLV vector and then were treated with the inhibitors 2 h after the inoculation. The treatment after inoculation much less efficiently attenuated CHIKV-pseudotyped MLV infection (Figure 5). This result suggests that the macropinocytosis inhibitors suppress an early step of pseudotyped vector infection, supporting the result that macropinocytosis is required for CHIKV E protein-mediated entry into host cells.

### 3.4. Cathepsin Protease Is Required for CHIKV-Pseudotyped MLV Vector Infection in 293T and TE671 Cells

MLV or Ebola virus infection is also mediated via acidic endosomes and requires endosomal cathepsin B protease. To assess whether cathepsin B was needed for the CHIKV-pseudotyped MLV infection, 293T and TE671 cells were pretreated with the cathepsin B inhibitor CA-074Me and then inoculated with the CHIKV-pseudotyped-MLV vector. We have already reported that CA-074Me treatment at the same concentration inhibits Ebola virus glycoprotein- or ecotropic MLV envelope protein-mediated infection without affecting the cell viability [18]. The CA-074Me treatment (20 or 40 μM) of 293T and TE671 cells inhibited the CHIKV-pseudotyped MLV vector infection in a dose-dependent manner (Figure 6A) but not the VSV-pseudotyped vector infection (Figure 6B). To confirm this result, 293T and TE671 cells were transduced by a lentiviral vector encoding cathepsin B (CatB) or a short hairpin RNA (shRNA) against CatB (shCatB). The cathepsin B mRNA level was analyzed by reverse transcriptase-PCR (RT-PCR). Cathepsin B mRNA expression was indeed reduced in the shCatB-expressing cells compared with that in the empty vector-transduced cells (Figure 6C). The exogenous expression of cathepsin B enhanced the infection by CHIKV-pseudotyped MLV vector in 293T and TE671 cells (Figure 6D), but slightly reduced the VSV-pseudotyped MLV vector infection (Figure 6E). The cathepsin B silencing decreased the relative infected cell numbers of the CHIKV-pseudotyped MLV vector, but the VSV-pseudotyped MLV vector infection was rather enhanced. This result demonstrated that cathepsin B is required for the CHIKV-pseudotyped vector infection.

To determine whether the CHIKV E protein was digested by cathepsin B protease, we first generated an antiserum against a peptide derived from the CHIKV E2 protein (Appendix A). Briefly, 293T cells were transfected with the CHIKV-pseudotyped MLV vector construction plasmids, and the vector particles were collected by centrifugation of the culture supernatants. Western immunoblotting of the CHIKV-pseudotyped MLV vector particles using the antiserum readily detected the mature E2 protein of the predicted molecular mass (Figure 6F). When the viral particles were treated with recombinant cathepsin B for 1 h at 37 °C, the amount of mature E2 protein was decreased and a novel 19 kDa peptide was observed. This result indicated that the CHIKV E2 protein was digested by cathepsin B protease.

## 4. Discussion

The results of this study suggest that endocytosis and macropinocytosis are necessary for the CHIKV-pseudotyped MLV vector infection in 293T and TE671 cells, respectively. Consistently, there are many lines of evidence showing that CHIKV enters host cells via endocytosis [8,9,10,11], and it has been recently reported that CHIVK entry is dependent on macropinocytosis in another rhabdomyosarcoma cell line [45]. The endocytosis inhibitors inhibited VSV-pseudotyped vector infection both in 293T and TE671 cells, suggesting that VSV-pseudotyped MLV vector infection is medicated by endocytosis. The viral entry route into the target cells may vary depending on the cell lines used and the environmental conditions, as we have already proposed, because endocytosis and macropinocytosis are complex processes controlled by many cellular factors [46]. To understand the mechanism, further study is needed.

This study also reveals that cathepsin B protease is important for the CHIKV-pseudotyped vector infection independent of the viral entry pathway. Whether the CHIKV-pseudotyped MLV vector particles are internalized into target cells by endocytosis or by macropinocytosis, cathepsin B protease is undoubtedly important for the infection. However, it is thought that only acidification is sufficient for activation of CHIKV E protein to induce membrane fusion [47]. CHIKV E2 protein digestion by cathepsin B protease may facilitate ability to promote membrane fusion. Cathepsin B overexpression may more efficiently activate the CHIKV E protein fusogenicity and enhance the infection than control cells. Alternatively, cathepsin B may digest host inhibitors against CHIKV-pseudotyped vector infection. This study also provides a possibility that cathepsin B inhibitors are potent therapeutic agents against CHIKV-induced diseases.

Cathepsin B silencing enhanced the VSV-pseudotyped MLV vector infection in both 293T and TE671 cells. This suggests that the endogenous cathepsin B protease probably inhibits VSV-pseudotyped vector infection by VSV-G protein digestion. We have previously reported that cathepsin B suppresses infection by CD4-independent strains of HIV [48]. Thus, we proposed that cathepsin proteases originally functioned as a host defense factor against viruses, and cathepsin-utilizing viruses appeared to have escaped the cathepsin protease-mediated host defense response [46].

The suppression of CHIKV-pseudotyped MLV vector infection by these inhibitors is not mediated by affecting autophagy. Autophagy participates for CHIKV replication [49,50]. Since lysosomal cathepsin proteiases are involved in LC3-II proteolysis, a maker for autophagy induction, the inhibitors used in this study should affect autophagy [51]. Thus, the inhibitors may suppress the CHIKV-pseudotyped vector infection by modulating autophagy. If so, the inhibitors should inhibit the infection both in 293T and TE671 cells. However, the endocytosis inhibitors attenuated the infection in 293T cells but not in TE671 cells, and the macropinocytosis inhibitors suppressed the infection in TE671 cells but not in 293T cells. Therefore, the inhibitors suppress CHIKV-pseudotyped vector infection by a mechanism other than affecting autophagy.

HeLa cells were much less susceptible to CHIKV-pseudotyped MLV vector infection than 293T and TE671 cells. However, replication-competent CHIKV efficiently proliferates in HeLa cells [52,53]. Thus, HeLa cells express the CHIKV receptor. CHIKV-pseudotyped vector infection was inhibited by the neutralizing antiserum, showing that the infection is mediated by the CHIKV E protein. HeLa cells may express an unknown host restriction factor that is counteracted by a CHIKV protein other than the E protein.

Survival of 293T cells was more severely affected by WOR than TE671 cells. However, another macropinocytosis inhibitor, EIPA, had a similar effect on cell viability of 293T and TE671 cells. Thus, it is unlikely that macropinocytosis plays an important role in 293T cell viability. WOR is an inhibitor of PI3K, and PI3K activation induces oncogenic cell growth [54]. Proliferation of 293T cells may require PI3K.

Human serum from a CHIKV-infected patient inhibited not only CHIKV-pseudotyped MLV vector infection but also VSV-pseudotyped MLV vector infection, although the inhibitory effect on CHIKV infection was higher than that on VSV infection. There are many lines of evidence showing that proteoglycans present in human serum inhibit MLV vector infection [55,56,57].

In summary, CHIKV E-containing viral particles are internalized into host cell vesicles by endocytosis or macropinocytosis, dependently on cell lines. By vesicle acidification, conformation of E protein is changed to activate its ability to promote membrane fusion. Although the conformational change of E protein by acidification is sufficient for infection, digestion of the E protein by endosomal cathepsin B protease more efficiently facilitates the membrane fusion activity and enhances infection.

## Figures and Tables

**Figure 1 viruses-12-00722-f001:**
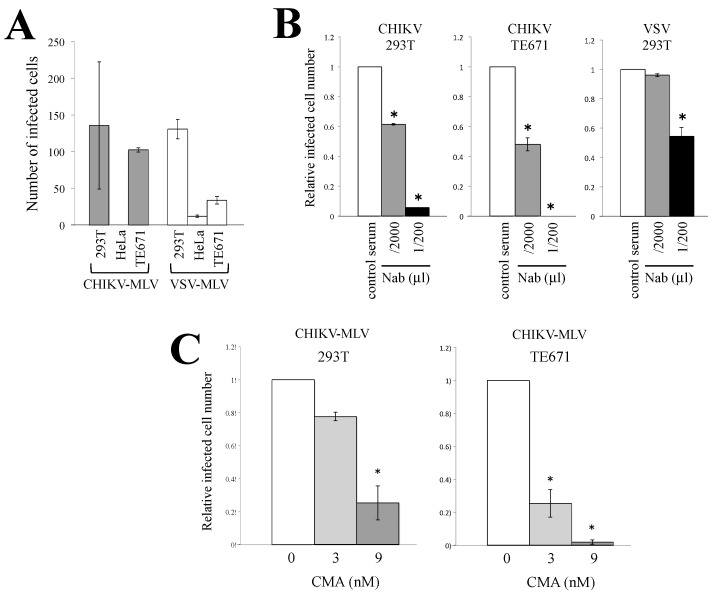
Endosome acidification is required for Chikungunya virus (CHIKV)-pseudotyped murine leukemia virus (MLV) vector infection. (**A**) 293T, HeLa, and TE671 cells were incubated with CHIKV-pseudotyped MLV vector. The cells were stained with X-Gal, and blue cells were counted in eight microscope fields. Total numbers of blue cells are indicated. This experiment was repeated in triplicate, and averages are shown. Error bars indicate standard deviations. (**B**) 293T and TE671 cells were inoculated with CHIKV-pseudotyped MLV vector in the presence of a serum from a CHIKV-infected patient that has neutralization activity, and numbers of blue cells were counted. Blue cell numbers in the presence of the control human serum were always set to 1. Relative values to the blue cell numbers detected in the cells treated with the same volume of dimethyl sulfoxide (DMSO) are indicated. This experiment was repeated in triplicate, and averages are shown. Error bars indicate standard deviations. Asterisks indicate statistically significant differences compared to the values in the presence of the control serum. (**C**) Concanamycin A (CMA)-pretreated 293T or TE671 cells were incubated with CHIKV-pseudotyped MLV vector, and the blue cells were counters after staining with trypan blue. Relative values to the blue cell numbers detected in the cells treated with the same volume of DMSO are indicated. Blue cell numbers in DMSO-treated cells were always set to 1. This experiment was repeated in triplicate, and averages are shown. Error bars indicate standard deviations. Asterisks indicate statistically significant differences compared to the values in DMSO-treated cells.

**Figure 2 viruses-12-00722-f002:**
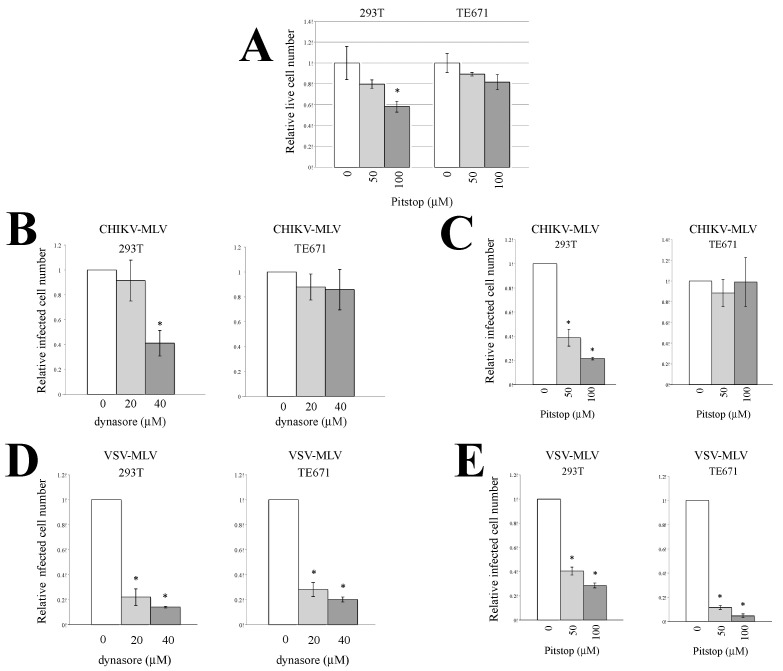
Endocytosis is required for CHIKV-pseudotyped MLV vector infection in 293T cells. (**A**) 293T and TE671 cells were treated with Pitstop2 for 5 h. The culture supernatants were changed to fresh medium without the inhibitors, and continued to be cultured for 24 h. Cell numbers were counted. This experiment was performed in triplicate. Relative values to numbers of cells treated with same volume of DMSO. Averages are shown. Error bars indicate standard deviations. Asterisks indicate statistically significant differences compared to the values in DMSO-treated cells. 293T or TE671 cells pretreated with the endocytosis inhibitor, dynasore (**A** and **C**) or Pitstop2 (**B**,**D**), were incubated with CHIKV (**B**,**C**)- or vesicular stomatitis virus (VSV) (**D** and **E**)-pseudotyped MLV vector. The cells were stained with X-Gal. Blue cell numbers in DMSO-treated cells were always set to 1. Relative values to the blue cell numbers detected in the cells treated with the same volume of DMSO are indicated. This experiment was repeated in triplicate, and averages are shown. Error bars indicate standard deviations. Asterisks indicate statistically significant differences compared to the values in DMSO-treated cells.

**Figure 3 viruses-12-00722-f003:**
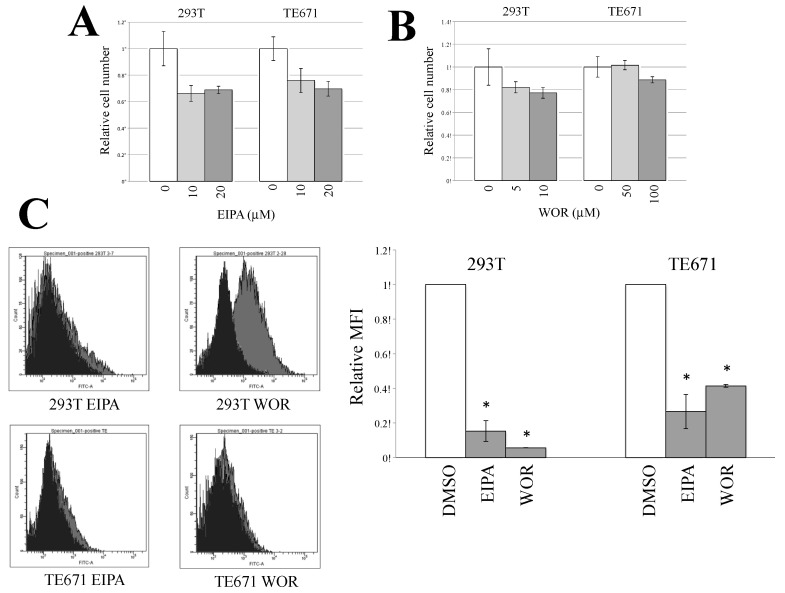
Effects of macropinocytosis inhibitors on cell viability and macropinocytosis. (**A** and **B**) 293T and TE671 cells were treated with 5-(N-ethyl-N-isopropyl)-amiloride (EIPA) or wortmannin (WOR) for 5 h. The culture supernatants were changed to fresh medium without the inhibitors, and continued to be cultured for 24 h. Cell numbers were counted. This experiment was performed in triplicate. Relative values to numbers of cells treated with same volume of DMSO. Averages are shown. Error bars indicate standard deviations. Asterisks indicate statistically significant differences compared to the values in DMSO-treated cells. (**C**) Fluorescein isothiocyanate (FITC)-conjugated dextran (molecular weight >70,000) internalized into cells by macropinocytosis was measured using a flow cytometer (left panel). Relative values to the means of fluorescence intensities (MFIs) in DMSO-treated cells are indicated (right panel). MFIs in the DMSO-treated cells are always set to 1. This experiment was repeated in triplicate, and averages are shown. Error bars indicate standard deviations. Asterisks indicate statistically significant differences compared to the values in DMSO-treated cells.

**Figure 4 viruses-12-00722-f004:**
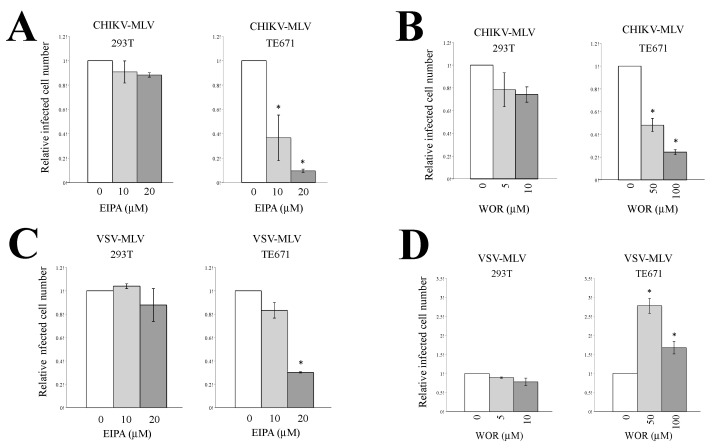
Macropinocytosis is required for CHIKV-pseudotyped MLV vector infection in TE671 cells. 293T or TE671 cells pretreated with the macropinocytosis inhibitor, EIPA (**A**,**C**), or WOR (**B**,**D**) were incybated with CHIKV (**A**,**B**)- or VSV (**C**,**D**)-pseudotyped MLV vector. The cells were stained with X-Gal. Relative values to the blue cell numbers detected in the cells treated with the same volume of DMSO are indicated. Blue cell numbers in DMSO-treated cells were always set to 1. This experiment was conducted in triplicate, and averages are shown. Error bars indicate standard deviations. Asterisks indicate statistically significant differences compared to the values in DMSO-treated cells.

**Figure 5 viruses-12-00722-f005:**
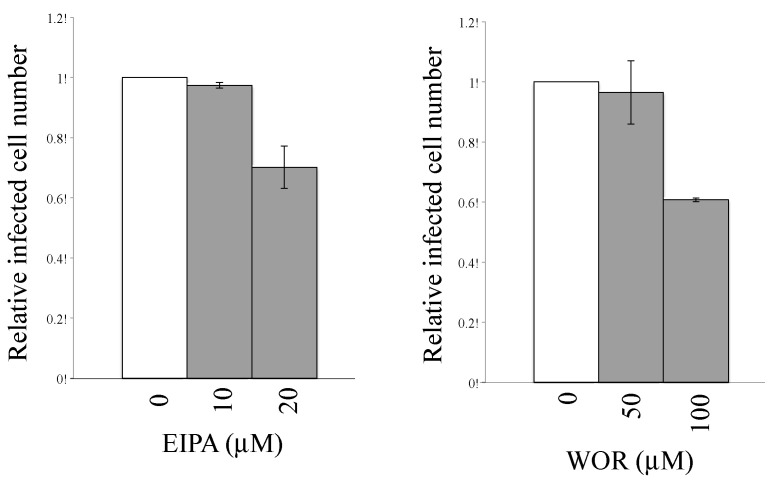
Treatment with macropinocytosis inhibitors after CHIKV-pseudotyped MLV vector inoculation inhibits its infection less efficiently than treatment before inoculation. TE671 cells were inoculated with CHIKV-pseudotyped MLV vector, and cultured for 5 h. The inoculated cells were treated with EIPA or WOR for 5 h. Blue cell numbers in DMSO-treated cells were always set to 1. Relative values to the blue cell numbers detected in the cells treated with the same volume of DMSO are indicated. This experiment was repeated in triplicate, and averages are shown. Error bars indicate standard deviations.

**Figure 6 viruses-12-00722-f006:**
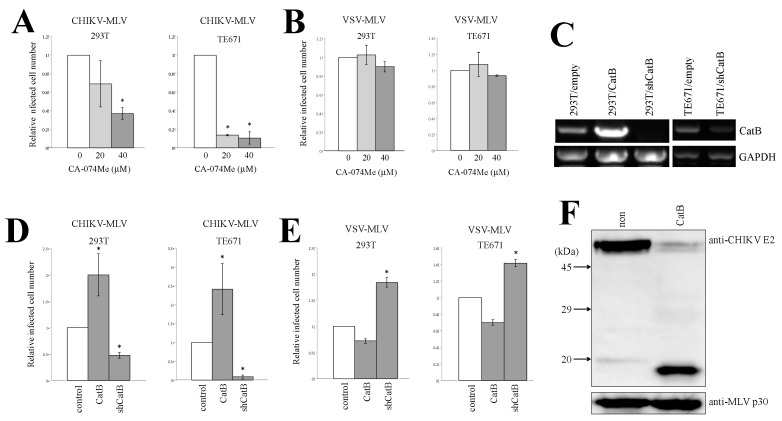
Cathepsin B is required for CHIKV-pseudotyped MLV vector infection in 293T and TE671 cells. 293T or TE671 cells pretreated with CA-074Me (**A**,**B**) or transduced by empty, cathepsin B, or shCatB-expressing lentiviral vector (**D**,**E**) were incubated with CHIKV (**A**,**D**)- and VSV (**B**,**E**)-pseudotyped MLV vector. The cells were stained with X-Gal. Relative values to the blue cell numbers detected in the cells treated with the same volume of DMSO are indicated. Blue cell numbers in DMSO-treated cells were always set to 1. Relative values to the blue cell numbers detected in the control cells are indicated. This experiment was repeated in triplicate, and averages are shown. Error bars indicate standard deviations. Asterisks indicate statistically significant differences compared to the values in DMSO-treated cells. mRNA levels of cathepsin B and glyceraldehyde-3-phosphate dehydrogenase (control) in 293T or TE671 cells transduced by the empty, cathepsin B, or shCatB-expressing lentiviral vector were analyzed by RT-PCR (**C**). CHIKV-pseudotyped MLV vector particles were treated with recombinant cathepsin B at 37 °C for 1 h and analyzed by western blotting (**F**).

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
