# Peer review of "Cathepsin B Protease Facilitates Chikungunya Virus Envelope Protein-Mediated Infection Via Endocytosis or Macropinocytosis"

_viruses, 2020, doi:10.3390/v12070722_

Round 1

Reviewer 1 Report

This paper from Izumida, et al. provides interesting data on the CHIK infectious process and the importance of cathepsin B's action on the E2 envelope protein of CHIKV to promote the membrane fusion step, necessary for the release of the virus from the endosome. The experiments based on pseudo typed MLV vectors mimicking CHIKV with CHIKV-E expressing viral particles also confirm that the CHIK envelope protein is associated with the attachment and entry steps during the infectious process. Although this data is already known, the authors confirm that the cell entry can take place via several pathways (here endocytosis and macropinocytosis) depending on the cell type.

Whilst these are potentially interesting observations, there are some very significant issues with the manuscript and some experiments that would need to be addressed before it would be considered to be suitable for publication. Many points need to be clarified to avoid misleading or confusing the reader.

Abstract

Right from the start, I'm embarrassed by the sentence. “Chikungunya virus (CHIKV) is an enveloped virus that enters host cells via endosomes”

It would be more accurate to say ‘Chikungunya virus (CHIKV) is an enveloped virus that enters host cells and transits within the endosomes’ before starting its replication cycle.

Introduction

There are few semantic issues and misstatements that are detrimental to comprehension and minor grammatical errors that also need fixing.

Line 42 : the clathrin-dependent endocytosis pathway involves clathrin-coated pits and vesicles but not « coated endosomes ». Please correct this statement.

Line 42-45 : The description of the entry processes of CHIKV is incomplete.  There is indeed work which has shown that phagocytosis inhibitors are also able to reduce infection and that recognition and entry may be dependent on exposed phosphatidyl-serine (perhaps via TIM) and apoptotic mimicry, with annexinV as effective as neutralizing antibodies in inhibiting the infection. This phagocytosis allows CHIKV to diversify its tropism and in particular to infect phagocytic cells (Krejbich-Trotot et al. Faseb 2010 https://doi.org/10.1096/fj.10-164178). In addition, opsonized viruses (with masked envelope proteins) can follow a Fcgamma-dependent phagocytosis pathway leading to antibody dependant enhancement. Please add this essential background information and references.

It’s ok that the finding remains that, regardless the way of entry, the infectious process involves a step in the endosomes in which acidification allows the release and uncoating of viral genomes in the cytoplasm.

Line 46 : inaccurate: there is a fusion between viral and endosomal membranes, not a protein E fusion!

Line 47. Requires reformulation and cautiousness. E proteins that emerge from the viral envelope may play an important role for recognition, attachment and entry step for several cell types but not for all and may be play a role at the post entry step when fusion is required between the endosomal and viral membranes to release the viral genetic material.

48 : rephrase. A polyprotein is not translated. What is translated is an RNA.

50 : reformulate for consistency with the viral cycle timing: 1) cell receptors 2) fusion

71-73 : poor statement: replace with 'infection by enveloped viruses'

and then ‘occur via endosome’ which is improper, replace by require a transit via the endosomal compartment.

80 : the choice of terms is tricky: co-stain ? be more specific.

Results

2.1  chikv pseudo typed retrovirus vector

 Please, provide a graphical summary of the rationale for the use of this vector, ahead of the results, describing the generation of pseudo typed MLV after a double transfection, the collect of the supernatants, the incubation of target cells and the measurement of the blue phenotype revealing intake and integration processes.

105, but also 112, 158 , 314 … and in the figure legends… ‘inoculated into’ is an inadequate sentence. This would suggest that you introduced the supernatants containing the pseudotyped MLVs ‘into’ the cells!

Cells are incubated with the chikv pseudo typed MLV containing supernatants!

This is a recurring mistake. Be sure to replace each time by more relevant terms: for example, « the cells have been put in contact with », or « incubated with », but please do not use this term ‘inoculated ‘which is not suitable in vitro.

107 and Figure 1A legend. Confusing. Please clearly define what you call a ‘relative titer’ (sometimes called transduction titer in the text). Especially since you are talking about blue cell counts but you have ones due to xgal and others due to trypan blue.

Again, « CHIKVE mediated infection » measure is not clearly explained. Blue cell counts are averages of 8 observed fields or of different independent experiments?

These measurements would need to be calibrated: how many cells have been brought into contact with the MLV vectors and does this correspond to a fixed MOI? Do the variations (huge sd) observed reflect the differential between the number of cells at the time of incubation and the number of infectious units?

A control with another vector, like MLV-VSV is missing.

An experiment with serial dilutions of the supernatant would be informative to get an idea of the infectious units efficiency.

We also don't understand if the value for HeLa cells is zero?

‘much higher’ is therefore an understatement. Is your guess that HeLa cells are not susceptible to CHIKVE-MLV ?

Yet, HeLa cells are described to be susceptible to ChikV infection. So is it possible to have infectious initiation steps that are totally E independent? Discuss.

110 : short-cuts used are wrong. Replace with “the infection by the pseudotype MLV vector is achieved by a recognition step dependent on ChikV E protein and we wanted to verify that this pseudotype virus was likely to be neutralized by an anti chik serum, as is CHIKV. Moreover, are the serum doses used compatible with sero-neutralization of CHIKV on the same cell type? This control experiment or a reference is missing.

115-116 and 190-192 : VSV-pseudotyped MLV vector infection / VSV-G-mediated infection. Define and homogenize this confusing nomenclature.

133 : ambiguity since you're not using the CHIKV virus: CHIKV E mediated infection, by the pseudotype MLV vector… Exactly as for the previous remark, try to set-up a single and straightforward terminology to which the reader can refer throughout the article.

2.2. Endocytosis is required for CHIKV E-mediated infection in 293T but not in TE671 cells

154 : replace insufficiently precise terms : dynasore inhibits vesicle formation

158: has to be changed because 40% mortality cannot be considered to be moderate.

Figure 2: 20 to 40% death with pitsop, that's huge in HEK compared to TE 671. Do you have an explanation?

2.3. Macropinocytosis is required for CHIKV E-mediated infection in TE671 cells 167 but not in 293T cells

 Figure 4 : homogenize the scales histogram 4D

190 192 : VSV-pseudotyped MLV vector infection /VSV-G-mediated infection

258 The endosome inhibitors inhibited VSV-G-mediated infection both in 293T and TE671 cells, suggesting that TE671 cells can induce endocytosis. 1) You can’t talk about “endosome inhibitors”. What is your data supporting that endosomes of the treated cells have disappeared? Show us microscopy images with the appropriate markers. 2) in TE 671 the VSV-pseudotyped MLV vector infection is mediated by endocytosis

270 not “membrane fusion capability” but ability to promote membrane fusion

284 285 where are the results you’re discussing here?

333 “construction” is not a suitable term for an antiserum. Need to be fixed

Author Response

Thank you very much for your valuable comments.

”Chikungunya virus (CHIKV) is an enveloped virus that enters host cells via endosomes”

            As the reviewer suggested, the sentence was changed to “Chikungunya virus (CHIKV) is an enveloped virus that enters host cells and transits within the endosomes before starting its replication cycle” (line 15-16).

>Line 42

            As the reviewer suggested, the sentence was changed to “clathrin-dependent endocytosis pathway” (line 45).

>Line 42-45

            As the reviewer suggested, the following sentences and references were added (line 48-52).

“In addition, it has been reported that phogocytosis inhibitors are able to reduce CHIKV infection [15] and antibody-bound viral particles can follow an Fcg-dependent phagocytosis pathway leading to antibody-dependent enhancement of viral infection [16], suggesting that CHIKV also enters to target cells via phagosomes.”

>Line 46

            As the reviewer suggested, the sentence was changed to “fusion between viral and endosome membrane” (line 47-48).

>Line 47

            As the reviewer suggested, the following sentences were added (line 56-58).

“The E proteins that emerge from the viral envelope play an important role for attachment to the cell receptor and entry step for many cell types, but not for all. The E proteins may also involve in the post entry step.”

>Line 48

            As the reviewer mentioned, the sentence was changed to “The viral protein is synthesized as a precursor polyprotein from a subgenomic 26s RNA” (line 53-54).

>Line 50

            As the reviewer suggested, the sentences were changed to “The E proteins that emerge from the viral envelope play an important role for attachment to the cell receptor and entry step for many cell types, but not for all. The E proteins may also involve in the post entry step.” (Line 56-58).

>Line 71-73

           As the reviewer suggested, the sentence was changed to “Infections by Ebola virus [17,18], SARS-CoV [19], SARS-CoV-2 [20], and murine leukemia virus (MLV) [18] also require a transit via the endosome compartment” (line 92-93).

>Line 80

            As the reviewer suggested, the sentence was changed to “skeletal muscle cells have found to contain CHIKV antigens” (line 101).

>Provide a graphical summary of the rationale for the use of this vector.

            As the reviewer suggested, the graphical summary of the pseudotyped vector was added (Figure S1).

>Line 105, 112, 158, 314....

            As the reviewer suggested, these sentences were changed to “Cells were incubated with the vector” throughout the manuscript.

>Line 107 and Figure 1A

            As the reviewer suggested, the words were changed to “infected cell number” throughout the manuscript.

>A control with another vector

            As the reviewer suggested, VSV-pseudotyped MLV vector was used (line 140-141)(Figure 1B).

>Serial dilution

            When the supernatant from the transfected cells was diluted with fresh medium by 1/2 and 1/4 times, numbers of infected cells were decreased to 1/2 and 1/4, respectively.

>The value for HeLa is zero?

         It is not zero. A sentence “Only a few infected cells were detected in HeLa cells.” is added (line 130-131). In discussion section, the following sentences were added (line 306-308).” CHIKV-pseudotyped vector infection was inhibited by the neutralizing antiserum, showing that the infection is mediated by the CHIKV E protein. HeLa cells may express an unknown host restriction factor that is counteracted by a CHIKV protein other than the E protein.”

>Line 110

            As the reviewer suggested, the sentence was changed to “We wanted to verify that the infection by the pseudotyped MLV vector is achieved by a recognition step dependent on CHIKV E protein and is neutralized by an anti-serum, as replication-competent CHIKV” (line 132-134).

>Line 115-116 and 190-192

            As the reviewer suggested, this nomenclature is homogenized as VSV-pseudotyped MLV vector infection throughout the manuscript.

>Line 154

            As the reviewer suggested, the sentence was changed to “Dynasore inhibits vesicle formation” (line 163-164).

>Line 158

            As the reviewer suggested, “moderately” was changed to “about 60%” (line 167).

>Figure 2

            As the reviewer suggested, the following sentences were added to the Discussion section (line 309-313). “Survival of 293T cells was more severely affected by WOR than TE671 cells. However, another macropinocytosis inhibitor, EIPA, had the similar effect on cell viability of 293T and TE671 cells. Thus, it is unlikely that macropinocytosis plays an important role in 293T cell viability. WOR is an inhibitor of PI3K, and PI3K activation induces oncogenic cell growth [51]. Proliferation of 293T cells may require PI3K.”

>Figure 4

            The scales of Figure 4D graphs are same.

>Line 190-192

            As the reviewer suggested, this nomenclature is homogenized as VSV-pseudotyped MLV vector infection throughout the manuscript.

>Line 258

            As the reviewer suggested, the sentence was changed to “The endocytosis inhibitors inhibited VSV-pseudotyped vector infection both in 293T and TE671 cells, suggesting that VSV-pseudotyped MLV vector infection is medicated by endocytosis” (line 271-273).

>Line 270

            As the reviewer suggested, the sentence was changed to “CHIKV E2 protein digestion by cathepsin B protease may facilitate ability to promote membrane fusion” (line 281-282).

>Line 284-285

            We added the result (Figure 1A).

>Line 333

  “Construction” was deleted (line 365).

Reviewer 2 Report

The authors have used a Pseudotyped viral technique to investigate the infection of CHIKV in 293T and TE671 cells. First, they demonstrated that endosomal acidification is important for viral infection because CMA attenuated the viral infection. Interestingly, they found that endocytosis is required for CHIKV infection in 293T cells but not in TE671 cells and that macropinocytosis is required for viral infection in TE671 cells but not 293T cells.

The data support the conclusion they made and I suggest accepting the manuscript.

Author Response

Thank you very much for accepting our manuscript.

Reviewer 3 Report

The manuscript is interesting and should be accepted for publication in Viruses.

However, I have a few comments on the manuscript as mentioned below.

It is well documented that differences do exist between (+)-strand RNA virus families such as hepatitis C virus (HCV), mosquito-borne flaviviruses, alphaviruses (e.g. Chikungunya virus), SARS CoVs, or non-enveloped picornaviruses in subtle ways. The pathogenesis of even members of the same family is often determined by the nature of the virus-host interaction.

Although evidence from previous studies indicate that lysosomal cathepsins are involved in LC3-II proteolysis and accumulation of LC3-II occurs when these lysosomal cathepsins are inhibited, supporting the hypothesis that cathepsin inactivation is the major cause of accumulation of LC3-II (Qiao, S., Tao, S., Rojo de la Vega, M., Park, S.L., Vonderfecht, A.A., Jacobs, S.L., Zhang, D.D., Wondrak, G.T., 2013).

However, many inhibitors are known to cause autophagic-lysosomal and proliferative blockade sensitizing human melanoma cells to starvation- and chemotherapy-induced cell death (Autophagy 9, 2087-2102). Other previously reported studies on drugs target the essential autophagy pathway involved in membrane biogenesis that is required for replication and assembly of (+)-strand RNA viruses. (Rubinsztein, D.C., Codogno, P., Levine, B., 2012. Autophagy modulation as a potential therapeutic target for diverse diseases. Nat. Rev. Drug Discov. 11, 709-730).

I was wondering whether the authors would like to comment on any role of autophagy in CHIKV.

Author Response

Thank you very much for your valuable comments.

       As the reviewer suggested, the following sentences about autophagy were added (line 294-302). “The suppression of CHIKV-pseudotyped MLV vector infection by these inhibitors is not mediated by affecting autophagy. Autophagy participates for CHIKV replication [46,47]. Since lysosomal cathepsin proteiases are involved in LC3-II proteolysis, a maker for autophagy induction, the inhibitors used in this study should affect autophagy [48]. Thus, the inhibitors may suppress the CHIKV-pseudotyped vector infection by modulating autophagy. If so, the inhibitors should inhibit the infection both in 293T and TE671 cells. However, the endocytosis inhibitors attenuated the infection in 293T cells but not in TE671 cells, and the macropinocytosis inhibitors suppressed the infection in TE671 cells but not in 293T cells. Therefore, the inhibitors suppress CHIKV-pseudotyped vector infection by a mechanism other than affecting autophagy.”

Round 2

Reviewer 1 Report

I consider that the authors have correctly addressed the remarks and questions I had expressed. Accordingly, and in view of the changes made to the manuscript, I approve the publication of the revised version.